# Early individualized risk prediction using clinical data for children during the febrile phase of dengue in outpatient settings in Vietnam and Thailand

Sorawat Sangkaew[1,2‡], Bethan Cracknell Daniels[3‡], Damien K. Ming[4], Bernard Hernandez[5], Pau Herrero[5,6], Piyarat Suntarattiwong[7], Siripen Kalayanarooj[7], Anon Srikiatkhachorn[8,9], Alan L. Rothman[9], Darunee Buddhari[10], Nguyen Lam Vuong[11,12], Phung Khanh Lam[11,12], Minh Tuan Nguyen[13], Bridget Wills[11,14], Cameron Simmons[15], Christl A. Donnelly[3,16], Sophie Yacoub[11,14‡], Alison Holmes[4,6‡], Ilaria Dorigatti[3‡*]

1 Department of Social Medicine, Hatyai Hospital, Songkhla, Thailand, 2 Section of Adult Infectious Disease, Department of Infectious Disease, Faculty of Medicine, Imperial College London, London, United Kingdom, 3 MRC Centre for Global Infectious Disease Analysis, Department of Infectious Disease Epidemiology, School of Public Health, Imperial College London, London, United Kingdom, 4 Centre for Antimicrobial Optimisation, Imperial College London, London, United Kingdom, 5 Department of Electrical and Electronic Engineering, Imperial College London, London, United Kingdom, 6 The David Price Evans Global Health and Infectious Diseases Unit, Faculty of Health & Life Sciences, University of Liverpool, Liverpool, United Kingdom, 7 Department of Pediatrics, Queen Sirikit National Institute of Child Health, Bangkok, Thailand, 8 Faculty of Medicine, King Mongkut's Institute of Technology Ladkrabang, Bangkok, Thailand, 9 Institute for Immunology and Informatics and Department of Cell and Molecular Biology, University of Rhode Island, Providence, Rhode Island, United States of America, 10 Department of Virology, WRAIR-AFRIMS, Bangkok, Thailand, 11 Oxford University Clinical Research Unit, Wellcome Trust Asia Programme, Ho Chi Minh City, Vietnam, 12 Faculty of Public Health, University of Medicine and Pharmacy at Ho Chi Minh City, Ho Chi Minh City, Vietnam, 13 Children's Hospital No. 1, Ho Chi Minh City, Vietnam, 14 Centre for Tropical Medicine and Global Health, University of Oxford, Oxford, United Kingdom, 15 Department of Microbiology and Immunology, Peter Doherty Institute for Infection and Immunity, University of Melbourne, Parkville, Melbourne, Australia, 16 Department of Statistics, University of Oxford, Oxford, United Kingdom,

‡ SY, AH and ID are joint senior authors on this work. SS and BCD are joint first author on this work.
* i.dorigatti@imperial.ac.uk

## Abstract

Dengue severity prediction models are usually developed using hospitalized patient data, but triage and hospital admission are mainly evaluated in outpatient settings. This study developed models using clinical and laboratory data from patients in outpatient settings during the febrile phase. Data from two cohort studies in Vietnam and Thailand were used to develop and validate six models: logistic regression with warning signs, Lasso-selected logistic regression, random forest, extreme gradient boosted classification, support vector machine, and artificial neural network. Models predicted dengue shock syndrome (DSS) as the primary endpoint and moderate plasma leakage and/or DSS as the secondary endpoint. We assessed model performance, discrimination, and calibration, using sensitivity, specificity, accuracy, Brier score, AUROC, CITL, calibration slope, calibration plots, and decision curve analysis.

**Data availability statement:** The data used in this study are not publicly available due to third-party data governance and institutional data-sharing policies. Data access requests can be submitted to the relevant institutions as follows. For AFRIMS data: Requests should be directed to the Chief, Department of Virology, WRAIR-AFRIMS (Email: John.Brooks.mil@afrims.org). Material has been reviewed by the Walter Reed Army Institute of Research. There is no objection to its presentation and/or publication. The opinions or assertions contained herein are the private views of the authors and are not to be construed as official, or as reflecting the true views of the Department of the Army or the Department of Defense. For OUCRU data: Requests can be submitted to the OUCRU Data Access Committee (DAC) via DAC@oucru.org. OUCRU recognizes the ethical obligation to ensure optimal use of the data it collects and to share individual-level data responsibly. In line with journal and sponsor regulations, OUCRU will upload the data supporting the findings of this study and the associated R code to the Oxford University Research Archive (ORA). Other data can be requested through the DAC. OUCRU's data sharing policy is available at: https://www.oucru.org/data-sharing-policy/ and the data request form at: https://www.oucru.org/wp-content/uploads/2023/06/OUCRU-Data-Request-Form-V1.1-090217.pdf The data are available from the corresponding institutions upon reasonable request and subject to applicable ethical and regulatory approvals.

**Funding:** The author(s) received no specific funding for this work.

**Competing interests:** The authors have declared that no competing interests exist.

The optimal model was the Lasso-selected logistic regression for predicting DSS and the combined endpoint of moderate plasma leakage and/or DSS (Brier score: 0.044 [95% CI 0.043, 0.044] and 0.104 [95% CI 0.104, 0.105]; AUROC: 0.789 [95% CI 0.787, 0.791] and 0.741 [95% CI 0.740, 0.742]). We identified hematocrit, platelet count, lymphocyte count, and aspartate aminotransferase as predictors for DSS, and abdominal pain or tenderness, vomiting, mucosal bleeding, white blood cell count, lymphocyte count, platelet count, aspartate aminotransferase, and serum albumin as predictors for the secondary endpoint. Logistic regression and machine learning models using clinical and laboratory data during the febrile phase can support early prediction of severe disease in outpatient settings. Integrating risk prediction models into a decision support system could improve triage and optimize healthcare and resource allocation in endemic and resource-limited areas.

## Author summary

Most dengue risk models are developed from hospitalized patient data, despite triage occurring in outpatient settings. Few studies have examined early outpatient predictors, and none have undergone external validation across countries. In this study, we developed and validated dengue risk prediction models using logistic regression and machine learning with outpatient data from Vietnam and Thailand. Our prior systematic review and expert consultation informed predictor selection. The models outperformed the WHO warning signs alone in predicting dengue shock syndrome and moderate plasma leakage, demonstrating better discrimination and calibration. Models incorporating four to eight routinely collected clinical parameters show promise for guiding early triage and improving care allocation, especially in resource-limited, dengue-endemic settings.

## Introduction

Dengue is a mosquito-borne viral disease that heavily burdens public health systems globally, with 3.83 billion (3.45–4.09) people currently living in areas suitable for dengue transmission [1]. In 2024, the number of dengue cases reached the highest level on record, with more than 14,000,000 cases reported globally [2]. However, estimates suggest the true burden was 77.8 million cases (95% CI 50.1–101.2 million) [3,4]. Globally, Asia has the greatest dengue burden, with ~70% of the global dengue burden in this region [1,5]. The annual incidence of dengue hospitalization in Vietnam and Thailand is estimated at 142 and 136 people per 100,000, respectively [3]. In 2023, the estimated direct medical cost per dengue case increased markedly with inpatient care, rising from USD 7.51 (outpatient) to USD 160.09 (inpatient) in Thailand and from USD 27.82 to USD 65.84 in Vietnam [6]. Thus, inpatient care is a primary driver of the global dengue economic burden [7] despite only 7% of cases

being treated in inpatient settings [3]. Around 1–5% of symptomatic cases develop severe clinical syndrome, typically on days 4–6 of illness, and can be fatal without prompt supportive therapy [8]. No antiviral treatments are available, and the vaccines developed and licensed to date have complex efficacy profiles and are expected to have modest impacts on hospitalizations [9]. Therefore, early recognition of patients at higher risk of severe disease, requiring close monitoring in hospitals or appropriate for outpatient care, is critical to improving case management and healthcare resource allocation.

The World Health Organization (WHO), in collaboration with the Special Program for Research and Training in Tropical Diseases, recommends using warning signs to help triage patients in the febrile phase by identifying those at higher risk of developing severe dengue [8]. The warning signs checklist shows high sensitivity but moderate specificity, potentially causing many unnecessary admissions [10–12]. Notably, the checklist does not provide an individualized prediction for the risk of severe dengue.

Clinical data-driven prediction models can estimate individual risk of severe disease, improving patient triage during the early febrile phase of dengue illness. Although several prediction models for dengue severity exist, most rely on hospitalized patient data [13–16], which may create a selection bias toward more severe presentations from the outset. Conversely, risk prediction models using data collected in the early febrile phase from outpatient settings can inform the triage and hospitalization admission of patients. As well as the early identification of high-risk individuals, this also reduces the unnecessary hospitalizations of low-risk patients, which is particularly important in resource-limited settings or during large dengue outbreaks, which can quickly overwhelm healthcare settings. We identified five studies that developed risk production models using outpatient data [17–21]. All studies showed moderate to high performance on internal training and validation sets; however, no studies included external validation on an independent dataset from a different country, limiting the generalizability of their findings.

This study fills this gap by developing statistical and machine learning models to predict progression to (i) dengue shock syndrome (DSS) and (ii) moderate plasma leakage and/or DSS using data from the febrile phase of outpatient illness. We train our model using data collected from a cohort study in Vietnam and validate it on an independent dataset collected in Thailand.

## Results

### Patient characteristics

The Vietnamese study enrolled 8,100 patients, of which 2,245 had laboratory-confirmed dengue, resulting in 45% (1,019) of patients being hospitalized, with the remaining 55% (1,226) of patients managed as outpatients (non-hospitalized) (S1 Fig). Among outpatients, 1,185 (96.66%) completed follow-up; among hospitalized patients, 110 (10.79%) developed DSS, and 185 (18.16%) developed moderate plasma leakage. (S1 Fig). In the Thai dataset, 1,210 patients were enrolled; 524 had laboratory-confirmed dengue, all hospitalized. Amongst these, 36 (6.87%) children developed DSS, and 182 (34.73%) developed moderate plasma leakage. (S1 Fig).

The Thai and Vietnamese patient characteristics with complete data on the day of enrolment are shown in S1–S2 Tables. In both datasets, the mean age was 8–9 years, and 54–56% of patients were male. Patients with DSS were hospitalized later in both Vietnam and Thailand. In the Vietnamese dataset, vomiting and abdominal pain/tenderness were associated with both DSS and the combined endpoint, and mucosal bleeding was associated with the combined endpoint. No significant symptom differences were found in the Thai dataset (S1–S2 Tables). Higher AST and lower platelet counts were associated with DSS and the combined endpoint in both datasets, but higher hematocrit and lower serum albumin levels were only significant in the Vietnamese dataset. Secondary dengue infection (defined as detecting at least one positive dengue-specific IgG on the febrile and convalescence samples) was associated with DSS and the combined endpoint in both countries, while dengue serotype differences were observed in Vietnam but not in Thailand (S1–S2 Tables).

## Selected predictors

Based on our previous systematic review and meta-analysis [22], and discussion with dengue experts, 12 and 11 candidate predictors for DSS and the combined endpoint were selected, respectively, comprising demographic information (age and nutritional status), signs and symptoms (vomiting, abdominal pain or tenderness, skin hemorrhage, mucosal bleeding), and laboratory data (hematocrit, platelet count, white blood cell count, lymphocyte count, AST, serum albumin) collected during the febrile phase (S3 Table). Hematocrit was excluded for the combined endpoints as it was part of the outcome definition. Using Lasso selection, we selected four predictors for DSS (hematocrit, platelet count, lymphocyte count, AST) and eight predictors for the combined endpoint (vomiting, mucosal bleeding, abdominal pain and/or tenderness, platelet count, white blood cell count, lymphocyte count, AST, serum albumin).

Table 1 shows the association of candidate predictors measured at enrolment with the two clinical endpoints. Vomiting (OR = 1.67 [95%CI 1.13, 2.47] and OR = 1.68 [95%CI 1.31, 2.15]) and abdominal pain (OR = 2.98 [95%CI 1.28, 6.09] and OR = 2.50 [95%CI 1.38, 4.34]) were associated with both DSS and the combined endpoint. Skin bleeding was associated with DSS (OR = 1.89 [95%CI 1.14, 3.02]), and mucosal bleeding was associated with the combined endpoint (OR = 1.94 [95%CI 1.16, 3.11]). During the febrile phase, higher hematocrit and AST, and lower platelet count, lymphocyte count, and serum albumin were significantly associated with DSS and the combined endpoint (Table 1).

## Predictive performance

We assessed the predictive performance of two regression models (the reference model: logistic regression model using the WHO warning signs parameters as predictors and logistic regression with Lasso selection) and four machine learning models developed using all candidate predictors (random forest, RF, extreme gradient boosted tree classification, XGB, support vector machine, SVM, and artificial neural network, ANN). Fig 1 summarizes the conceptual framework of model development and internal validation, including 10-fold cross-validation to tune hyperparameters and 10-fold calibration to calibrate the model. Model training and validation were repeated 45 times to obtain mean and 95% CI estimates of predictive performance using Block Jackknife estimation. Predictive performance was assessed using the Brier score (a strictly proper scoring rule which measures the accuracy of probabilistic predictions), a discrimination measurement (the area under receiver operating characteristic curves, AUROC), calibration measurements (calibration plots, calibration in

**Table 1. Mean and 95% confidence interval (CI) of the Odds Ratio of developing dengue shock syndrome (DSS) and the combined endpoint (moderate plasma leakage and/or DSS) for each candidate predictor derived from univariate logistic regression.**

| Parameters | DSS | | Combined endpoint | |
|---|---|---|---|---|
| | Odds ratio (95%CI) | p-value | Odds ratio | p-value |
| Age (years) | 1.00 (0.95, 1.06) | 0.910 | 1.02 (0.99, 1.06) | 0.187 |
| Nutritional status: Overweight | 1.50 (0.87, 2.45) | 0.120 | 0.92 (0.62, 1.32) | 0.662 |
| Vomiting: Yes | 1.67 (1.13, 2.47) | 0.010 | 1.68 (1.31, 2.15) | <0.001 |
| Skin bleeding/positive tourniquet: Yes | 1.89 (1.14, 3.02) | 0.010 | 1.38 (0.97, 1.94) | 0.064 |
| Mucosal bleeding: Yes | 1.52 (0.62, 3.13) | 0.305 | 1.94 (1.16, 3.11) | 0.008 |
| Abdominal pain or tenderness: Yes | 2.98 (1.28, 6.09) | 0.005 | 2.50 (1.38, 4.34) | 0.002 |
| Hematocrit (%) | 1.14 (1.08, 1.20) | <0.001 | – | – |
| White blood cell count (1,000 cells/dl) | 0.94 (0.88, 1.01) | 0.127 | 0.96 (0.92, 1.00) | 0.061 |
| Platelet count (1,000 cells/dl) | 0.985 (0.981, 0.988) | <0.001 | 0.991 (0.988, 0.993) | <0.001 |
| Lymphocyte counts (1,000 cells/dl) | 0.69 (0.50, 0.92) | 0.017 | 0.52 (0.42, 0.65) | <0.001 |
| Serum albumin (g/L) | 0.91 (0.86, 0.97) | 0.003 | 0.89 (0.86, 0.93) | <0.001 |
| AST (units/dl) | 1.009 (1.007, 1.0113) | <0.001 | 1.007 (1.005, 1.009) | <0.001 |

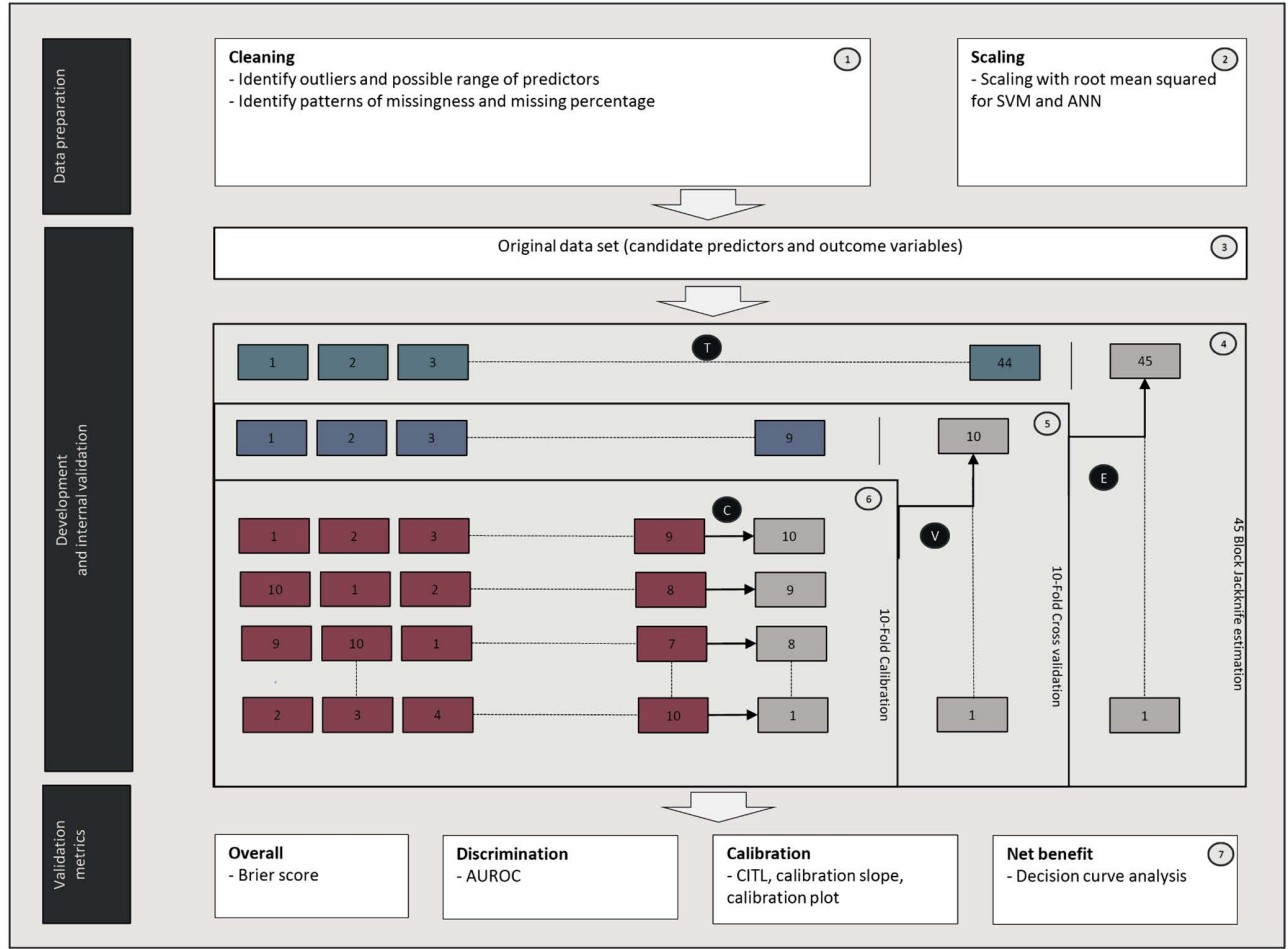

**Fig 1. Description of the workflow used for model development and internal validation.** After data preparation (Steps 1 and 2), the data are used to develop risk prediction models (Step 3). The dataset is split into 45 random blocks stratified by the outcome (Step 4). Each block (green boxes) is then used for hyperparameter optimization (Process T). The model is trained and validated on 44 of the 45 blocks with 10-fold cross-validation (Process V), leaving one block out in turn (Step 5). The training set (blue boxes) is divided into ten folds (in the Calibration loop): nine folds (red boxes) are used to develop the model, and the other fold (grey boxes) is used for Platt model calibration (Process C). Outputs from the calibrated model are the predicted probability of developing the endpoint (DSS in the primary analysis; moderate plasma leakage and/or DSS in the secondary analysis). The 10-fold validation and 10-fold calibration were repeated 45 times, each time leaving a different block and using the optimized hyper-parameterization. The central estimates and variance of the predictive performances are estimated from the 45 blocks using Block Jackknife estimation (Process E).

the large, CITL, and calibration slope), decision curve analysis (quantifies the clinical net benefit of models across a range of threshold probabilities for decision-making), the sensitivity, specificity, positive predictive value (PPV), and negative predictive value (NPV).

The predictive performance of the models developed on the Vietnamese dataset and internally validated using 10-fold cross-validation is presented in Table 2. The optimal hyperparameters and the coefficients of the multivariable logistic regression models with Lasso selection are presented in S4–S8 Tables. When predicting DSS, the reference model (logistic regression with the WHO warning signs) achieved the best overall and discrimination performance with an average accuracy of 0.756 [095% CI 0.752, 0.761], Brier score of 0.041 [95% CI 0.040, 0.041], and AUROC of 0.789 [95% CI 0.787, 0.791] (Table 2). Logistic regression with Lasso selection achieved better calibration performance (CITL -0.001

**Table 2. Predictive performance of the risk prediction models for DSS and the combined endpoint of moderate plasma leakage and/or DSS using multivariable logistic regression with the WHO warning signs as predictors, logistic regression with variables selected by Lasso selection, random forest, extreme gradience boosted tree, support vector machine, and artificial neural network with 2 hidden layers, all trained on the Vietnamese dataset on internal validation using 10-fold cross validation.**

| Model | Brier score (95% CI) *(the lower, the better)* | AUROC (95% CI) | Calibration intercept (95% CI) *(the closer to 0, the better)* | Calibration slope (95% CI) *(the closer to 1, the better)* | Accuracy (95% CI) | Sensitivity (95% CI) | Specificity (95% CI) | PPV (95% CI) | NPV (95% CI) |
|---|---|---|---|---|---|---|---|---|---|
| **Outcome: DSS** | | | | | | | | | |
| Logistic regression using WHO warning signs | 0.041 (0.040, 0.041) | 0.789 (0.787, 0.791) | -0.003 (-0.005, -0.001) | 1.027 (1.011, 1.043) | 0.756 (0.752, 0.761) | 0.779 (0.773, 0.785) | 0.755 (0.750, 0.76) | 0.154 (0.150, 0.158) | 0.985 (0.985, 0.986) |
| Logistic regression with variables selected by Lasso selection | 0.044 (0.043, 0.044) | 0.789 (0.787, 0.791) | -0.001 (-0.002, 0.001) | 0.942 (0.933, 0.951) | 0.731 (0.725, 0.736) | 0.780 (0.774, 0.787) | 0.728 (0.722, 0.734) | 0.138 (0.135, 0.142) | 0.985 (0.984, 0.985) |
| Random forest | 0.044 (0.044, 0.044) | 0.755 (0.753, 0.757) | -0.007 (-0.009, -0.004) | 1.026 (1.013, 1.038) | 0.715 (0.710, 0.720) | 0.771 (0.764, 0.779) | 0.712 (0.706, 0.717) | 0.128 (0.127, 0.130) | 0.984 (0.983, 0.984) |
| Extreme gradient boosted tree | 0.044 (0.044, 0.044) | 0.785 (0.783, 0.787) | -0.018 (-0.022, -0.015) | 1.053 (1.037, 1.070) | 0.749 (0.743, 0.755) | 0.751 (0.744, 0.758) | 0.749 (0.742, 0.755) | 0.146 (0.143, 0.149) | 0.983 (0.983, 0.984) |
| Support vector machine | 0.047 (0.047, 0.047) | 0.613 (0.607, 0.619) | -0.016 (-0.020, -0.012) | 0.404 (0.238, 0.57) | 0.756 (0.752, 0.761) | 0.779 (0.773, 0.785) | 0.755 (0.75, 0.76) | 0.154 (0.15, 0.158) | 0.985 (0.985, 0.986) |
| Artificial neural network:2 hidden layers | 0.045 (0.044, 0.045) | 0.775 (0.771, 0.779) | -0.047 (-0.056, -0.039) | 1.742 (1.598, 1.886) | 0.731 (0.725, 0.736) | 0.780 (0.774, 0.787) | 0.728 (0.722, 0.734) | 0.138 (0.135, 0.142) | 0.985 (0.984, 0.985) |
| **Outcome: combined endpoint of moderate plasma leakage and/or DSS** | | | | | | | | | |
| Logistic regression using WHO warning signs | 0.101 (0.101, 0.101) | 0.697 (0.696, 0.698) | -0.002 (-0.003, -0.001) | 0.965 (0.957, 0.973) | 0.684 (0.68, 0.689) | 0.661 (0.656, 0.667) | 0.688 (0.682, 0.693) | 0.254 (0.251, 0.257) | 0.930 (0.929, 0.931) |
| Logistic regression using variables selected by Lasso selection | 0.104 (0.104, 0.105) | 0.741 (0.740, 0.742) | 0.001 (0.001, 0.001) | 0.968 (0.962, 0.974) | 0.703 (0.700, 0.707) | 0.706 (0.701, 0.711) | 0.703 (0.698, 0.707) | 0.273 (0.270, 0.275) | 0.94 (0.939, 0.941) |
| Random forest (RF) | 0.104 (0.104, 0.105) | 0.735 (0.734, 0.736) | -0.004 (-0.005, -0.003) | 1.035 (1.027, 1.043) | 0.716 (0.712, 0.72) | 0.693 (0.687, 0.699) | 0.720 (0.714, 0.725) | 0.281 (0.278, 0.285) | 0.939 (0.938, 0.94) |
| Extreme gradient boosted tree (XGB) | 0.104 (0.104, 0.104) | 0.745 (0.744, 0.747) | -0.005 (-0.007, -0.002) | 1.040 (1.031, 1.050) | 0.721 (0.717, 0.724) | 0.696 (0.692, 0.701) | 0.724 (0.72, 0.729) | 0.289 (0.285, 0.292) | 0.940 (0.939, 0.94) |
| Support vector machine (SVM) | 0.111 (0.110, 0.111) | 0.673 (0.671, 0.676) | -0.009 (-0.011, -0.007) | 1.078 (1.056, 1.099) | 0.678 (0.673, 0.684) | 0.637 (0.631, 0.644) | 0.685 (0.678, 0.692) | 0.245 (0.241, 0.249) | 0.925 (0.924, 0.926) |
| Artificial neural network: 2 hidden layers (ANN) | 0.107 (0.107, 0.108) | 0.724 (0.722, 0.727) | -0.01 (-0.013, -0.007) | 1.31 (1.264, 1.355) | 0.691 (0.684, 0.697) | 0.689 (0.683, 0.696) | 0.691 (0.683, 0.699) | 0.26 (0.256, 0.264) | 0.935 (0.934, 0.936) |

AUROC: area under receiver operating characteristic curves; CI: Confidence interval; PPV: Positive predictive value; NPV: Negative predictive value.

[95% CI -0.002, 0.001], calibration slope 0.942 [95% CI 0.933, 0.951]) with comparable discrimination performance and AUROC (Table 2). Amongst the machine learning models, XGB performed best in terms of discrimination and calibration (Table 2). The optimal hyperparameters selected for the machine learning models are presented in S9 Table.

When predicting the combined endpoint, model performance was generally lower compared to DSS. The reference model (logistic regression using WHO warning signs) again obtained the best Brier score; however, its AUROC was lower than the logistic regression with Lasso selection (0.697 [95% CI 0.696, 0.698] versus 0.741 [95% CI 0.740, 0.742]). This reflects the higher sensitivity and specificity of the Lasso model in predicting the combined endpoint (Table 2). Overall, XGB achieved the highest discrimination (AUROC 0.745 [95% CI 0.744, 0.747]) and accuracy (0.721 [95% CI 0.717, 0.724]), but its calibration performance (CITL -0.005 [95% CI -0.007, -0.002], calibration slope 1.040 [95% CI 1.031, 1.050]) was worse than logistic regression with Lasso selection which showed the best overall calibration (CITL 0.001 [95% CI 0.001, 0.001], calibration slope 0.968 [95% CI 0.962, 0.974]) (Fig 2 and Table 2). RF and SVM underperformed in overall performance, discrimination, and calibration (Table 2).

Sensitivity, specificity, and accuracy for all models are presented in Table 2. For DSS, the Lasso model achieved a sensitivity of 0.780 (95% CI: 0.774–0.787), specificity of 0.728 (95% CI: 0.722–0.734), and accuracy of 0.731 (95% CI: 0.725–0.736), which were comparable to the WHO warning signs model (sensitivity 0.779 [95% CI: 0.773–0.785],

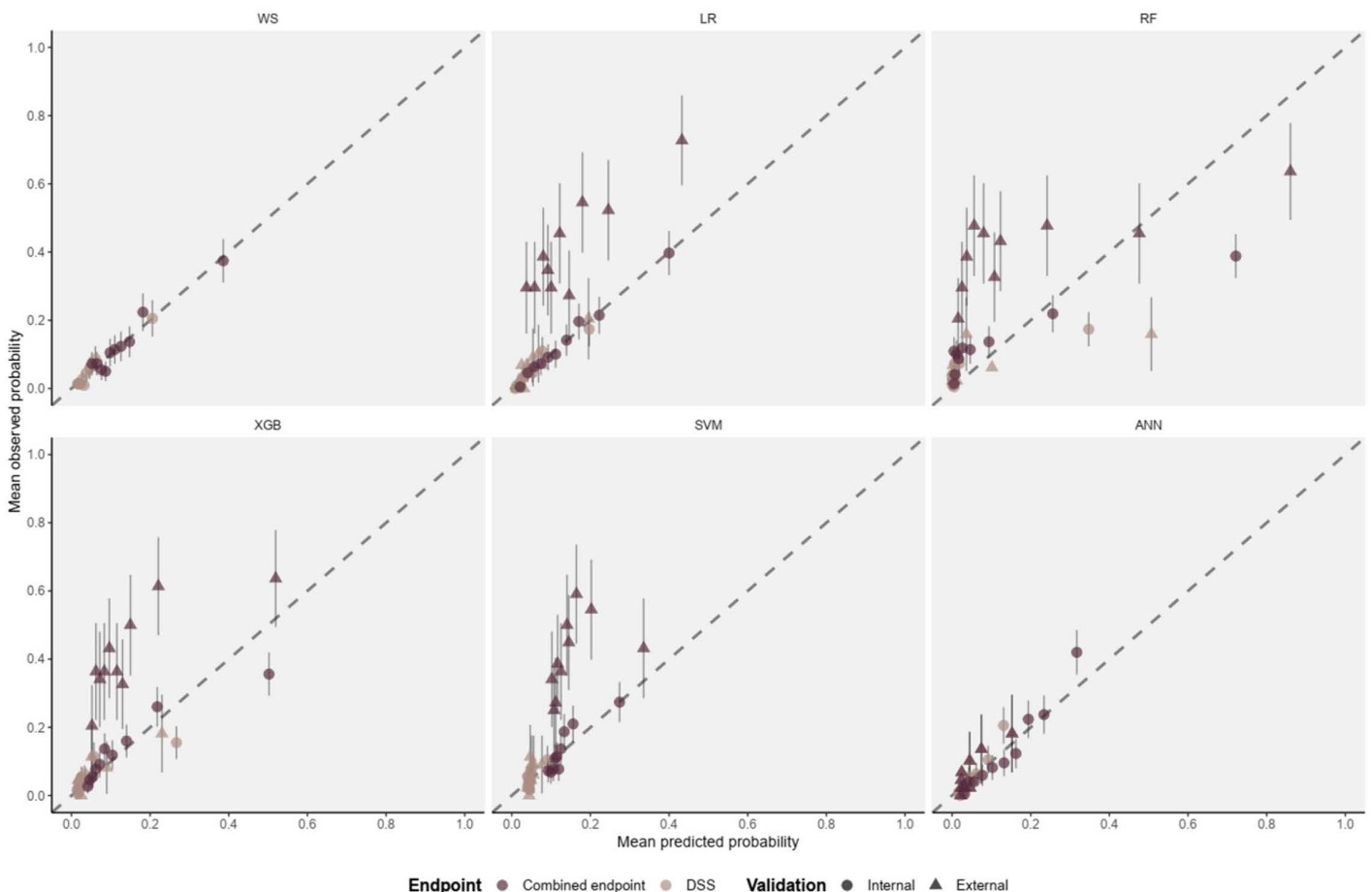

**Fig 2. Calibration plots showing the mean predicted probability (x-axis) versus the mean observed probability (y-axis) for DSS (triangles) and the combined endpoint of moderate plasma leakage and/or DSS (points) using multivariable logistic regression with the WHO warning signs as predictors (WS) (validated only internally), logistic regression with variables selected by Lasso selection (LR), random forest (RF), extreme gradience boosted tree (XGB), support vector machine (SVM), and artificial neural network with 2 hidden layers (ANN).** The diagonal dotted line in each panel represents the perfect agreement between predicted and observed risk. DSS: dengue shock syndrome.

specificity 0.755 [95% CI: 0.750–0.760], accuracy 0.756 [95% CI: 0.752–0.761]). For the combined endpoint of moderate plasma leakage or DSS, the Lasso model performed better, with sensitivity 0.706 (95% CI: 0.701–0.711), specificity 0.703 (95% CI: 0.698–0.707), and accuracy 0.703 (95% CI: 0.700–0.707). Similar patterns were observed across the machine-learning models.

Including days of illness (DOI) as an additional predictor did not improve model performance (S10 Table). For DSS, the AUROC was 0.789 (95% CI: 0.787–0.791) with the Lasso model and 0.778 (95% CI: 0.704–0.853) when DOI was included, with calibration and classification metrics remaining stable. Similar findings were observed for the combined endpoint of plasma leakage or DSS (AUROC 0.741 [95% CI: 0.740–0.742] without DOI vs. 0.732 [95% CI: 0.693–0.771] with DOI). In a subgroup analysis of hospitalized cases with the combined endpoint, logistic regression with Lasso selection achieved the highest performance (AUROC 0.648 [95%CI 0.647, 0.650], CITL 0.000 [95%CI -0.001, 0.001], calibration slope 0.903 [95%CI 0.892, 0.914]) (S11 Table).

Fig 3 visualizes the decision curve analysis showing the net benefit at a given risk threshold (probability of positive diagnosis). At a low threshold probability (< 0.1), the ANN model provided the highest net benefit for predicting DSS. Overall, however, the combined endpoint was more clinically valuable, with logistic regression with Lasso selection, XGB, and ANN achieving higher net benefits than logistic regression with the WHO warning signs. However, at higher threshold probabilities (>25%), logistic regression with the WHO warning signs had a higher net benefit. RF and SVM performed the worst across all thresholds (Fig 3).

## External validation

The Vietnamese (training) and Thai (external validation) datasets had different case mixes (membership model concordance statistic of 0.84 [95% CI: 0.81–0.86]). Differences in predicted risk further support these dataset differences (S2

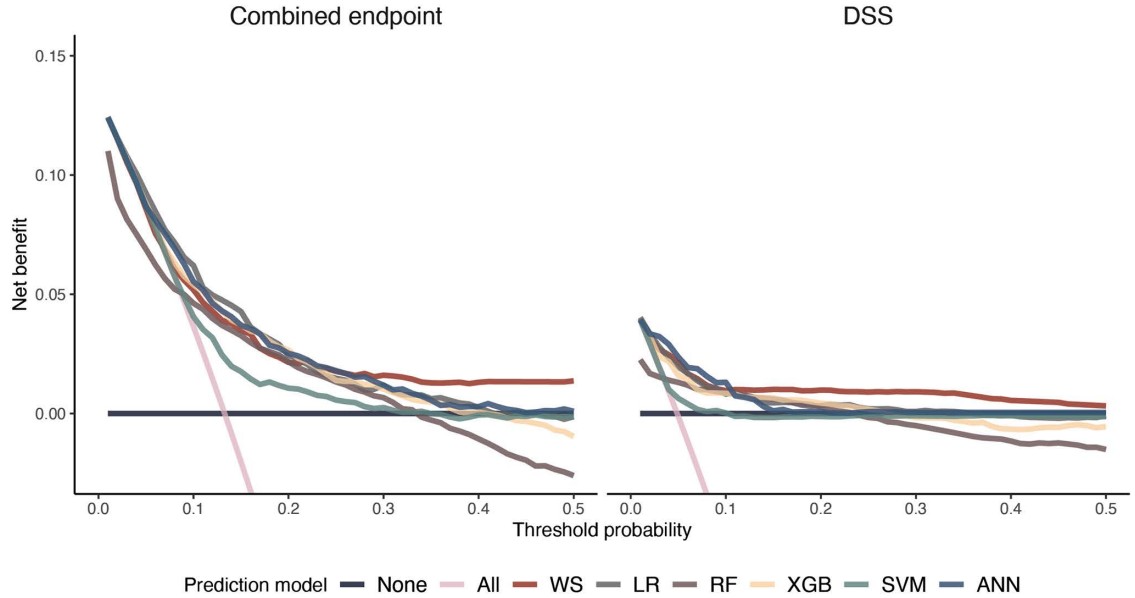

**Fig 3. Decision curves, showing the net benefit (y-axis) against the threshold probability (x-axis) for predicting moderate plasma leakage and/or dengue shock syndrome (DSS) (left panel) and DSS alone (right panel).** Models evaluated are logistic regression using the WHO warning signs (WS), logistic regression with variables selected by Lasso selection (LR), random forest (RF), Extreme gradience boosted tree (XGB), support vector machine (SVM), and artificial neural network with 2 hidden layers (ANN). All (pink diagonal line) represents treating all patients. None (grey horizontal line) represents treating no patients.

Fig). Predictive performance on the external validation dataset was lower than on the internal validation sets, with AUROC values for DSS and the combined endpoints 11% and 17% lower on average, respectively (Tables 2–3). Calibration metrics suggest prediction of both endpoints is generally underestimated (CITL higher than 0) and too extreme (calibration slope lower than 1) (Table 3 and Fig 2). The artificial neural network achieved the highest AUROC for DSS (0.690), but calibration remained imperfect (intercept 0.331, slope 0.912), consistent with the pattern observed in internal validation. By comparison, logistic regression with Lasso selection (AUROC 0.677, intercept 0.188, slope 0.531) and XGBoost (AUROC 0.679, intercept 0.248, slope 0.544) showed slightly lower discrimination but more stable calibration (Table 3). In the Thai cohort, application of the Lasso model reduced false positives compared with WHO warning signs (13% vs. 47% for DSS; 41% vs. 76% for plasma leakage), but at the cost of higher false negatives (26% vs. 6% for DSS; 2% vs. 0% for plasma leakage) (S12 Table). When predicted probabilities of DSS and plasma leakage were plotted against each predictor and stratified by country, the patterns of association were consistent across the Vietnamese and Thai datasets, suggesting that differences in baseline patient characteristics did not materially alter predictor–outcome relationships (S5–S6 Figs).

## Sensitivity analysis

In a sensitivity analysis, we optimized the classification threshold by considering the cost of false negatives and DSS prevalence for the logistic regression with Lasso selection and XGB models. We found that a cost value of approximately 20 maintains the accuracy level observed in the baseline scenario, while increasing the cost beyond 20 improves sensitivity, with the logistic regression with Lasso selection model identifying 90.1% of DSS cases at a cost of 100, while excluding 61.0% of non-DSS cases (S13 Table).

## Discussion

In outpatient settings, effective triage is key for identifying patients at risk of severe dengue. We developed risk prediction models using data from outpatient departments in Vietnam and Thailand to identify clinical and laboratory predictors

**Table 3. Predictive performance of the risk prediction models for DSS and the combined endpoint of moderate plasma leakage and/or DSS using multivariable logistic regression with the WHO warning signs as predictors, logistic regression with variables selected by Lasso selection, random forest, extreme gradience boosted tree, support vector machine, and artificial neural network with 2 hidden layers, on the Vietnamese (training) dataset and validated on the Thai (external validation) dataset.**

| Model | Fitted to Vietnamese dataset and externally validated on the Thai dataset | | | |
|---|---|---|---|---|
| | Brier score *(the lower, the better)* | AUROC | Calibration intercept *(the closer to 0, the better)* | Calibration slope *(the closer to 1, the better)* |
| **Outcome: DSS** | | | | |
| Logistic regression with Lasso selection | 0.059 | 0.677 | 0.188 | 0.531 |
| Random forest | 0.085 | 0.665 | -0.234 | 0.223 |
| Extreme gradient boosted tree | 0.060 | 0.679 | 0.248 | 0.544 |
| Support vector machine | 0.057 | 0.648 | 0.234 | 1.626 |
| Artificial neural network: 2 hidden layers | 0.057 | 0.690 | 0.331 | 0.912 |
| **Outcome: combined endpoint of moderate plasma leakage and/or DSS** | | | | |
| Logistic regression with Lasso selection | 0.299 | 0.624 | 1.636 | 0.287 |
| Random forest | 0.312 | 0.634 | 1.903 | 0.214 |
| Extreme gradient-boosted tree | 0.300 | 0.650 | 1.613 | 0.731 |
| Support vector machine | 0.303 | 0.572 | 1.352 | 0.391 |
| Artificial neural network: 2 hidden layers | 0.289 | 0.615 | 1.345 | 0.617 |

AUROC: area under the receiver operating characteristic curves; CI: Confidence interval.

collected during the febrile phases of illness. Internal validation showed that logistic regression with predictors selected by Lasso achieved the most balanced performance, combining good discrimination and calibration. Among machine-learning methods, XGBoost and ANN performed well, with XGBoost showing the most consistent and optimal results, particularly in external validation.

Unlike most previous studies [13–16], our models use predictors collected in outpatient settings during the febrile phase, providing earlier estimates of DSS risk to improve triage and patient management. Of the five studies previously predicting risk of severe dengue using outpatient data, two included no validation [18,19], and three performed model validation by splitting their dataset, either temporally or based on missing data [17,20,21]. Thus, by training our model on a Vietnamese dataset and validating it on an external Thai dataset, ours is the first to demonstrate the generalizability of a model trained on outpatient data across independent settings. We found good predictive performance on the external validation dataset (maximum AUROC 0.69) despite differences between the training and validation sets in transmission setting, hospital admission protocol, clinical management, and monitoring frequency.

To our knowledge, ours is also the first study using outpatient data to include a decision-curve analysis, allowing us to evaluate the net clinical benefit of our prediction models compared to hospitalization of all or none of the patients presenting at an outpatient setting, ensuring the models improve patient outcomes. We showed that at low threshold probabilities, machine learning models (ANN) offered the greatest net benefit in predicting DSS.

Given DSS is rare, we additionally considered a combined endpoint of moderate plasma leakage and/or DSS, increasing the PPV from 0.138 [95%CI 0.135, 0.142] to 0.273 [95%CI 0.270, 0.275] using the Lasso model. This combined endpoint also had a higher PPV compared to a similar study predicting DSS (PPV 0.10 [95%CI 0.09, 0.12]), reducing unnecessary inpatient care [21]. However, failing to identify individuals progressing to severe dengue poses a greater risk. In a sensitivity analysis, we showed how DSS prevalence and the cost of failing to identify at-risk individuals can be incorporated into the prediction model, tailoring it to individual settings and prioritizing DSS identification. All predictors included in this study are conventional laboratory parameters widely available in outpatient settings. They were carefully chosen based on our previous systematic review, meta-analysis [22], and in discussions with dengue experts. Using multivariable logistic regression and Lasso selection, we found four variables (hematocrit levels, platelet count, lymphocyte count, AST) and eight variables (vomiting, abdominal pain, mucosal bleeding, platelet count, white blood cell count, lymphocyte count, AST, serum albumin levels) were predictors of DSS and the combined endpoint, respectively. Platelet count and AST were consistently strong predictors in all models, in agreement with our previous meta-analysis [22], supporting testing and monitoring of these during the febrile phase. Our results also agree with vomiting, mucosal bleeding, and abdominal pain as established risk factors for developing severe dengue, which are warning signs in the 2009 WHO Dengue guidelines [8,22].

In this analysis, we investigated the association of severe dengue with the presence or absence of these symptoms, but the frequency and amount of vomiting, mucosal bleeding, and abdominal pain can also indicate the severity of the disease. In future work, it will be relevant to explore whether quantitative differences in vomiting, mucosal bleeding, and abdominal pain improve the prediction of DSS and/or plasma leakage. Additionally, we only considered the risk of severe dengue, whereas previous work has identified clinical fluid accumulation and elevated hematocrit as significant predictors of time taken to severe dengue progression in a hospitalized pediatric population in India [23]. Future work using Cox regression for semi-parametric analysis or discrete-time hazard models that leverage machine learning approaches to predict time to severe dengue progression could provide clinically actionable predictions for optimizing monitoring intensity, timing interventions, and allocating intensive care resources based on individual patient trajectories.

This study has additional limitations. The combined endpoint definition assumed non-hospitalized Vietnamese patients did not develop moderate plasma leakage unless detected during outpatient visits, potentially leading to underestimation. Limited chest X-rays for detecting plasma leakage might have caused missed cases. A higher hemoconcentration cut-off (e.g., 20% instead of 15%) would reduce under-reporting and the frequency of plasma leakage, affecting the relationship

between predictors and the combined outcome. The cut-off threshold for plasma leakage also influences the relatedness between the Vietnamese (training) and Thai (external validation) datasets. Reciprocal validation using the Thai dataset was not feasible because the limited number of outcome events would not have supported robust model development. Finally, our Thai and Vietnamese cohorts (1994–2008 and 2010–2013, respectively) may have limited contemporary transportability given temporal changes in dengue patient characteristics, including demographic shifts toward older patients, increasing obesity rates, and higher baseline liver enzyme levels in the region [24–27]. Prospective validation of our models in contemporary cohorts is therefore essential. The analysis offers new insights into predictive variables associated with severe dengue progression from two independent cohort studies in Thailand and Vietnam. It provides evidence that monitoring platelet count and AST during the febrile phase is useful for predicting severe disease. The models developed in this study are a step towards a decision support system for triage and clinical decision-making. While further testing and validation are needed, this work lays the foundation for integrating evidence-based, data-driven methods into a decision support system, optimizing healthcare allocation and resource use.

## Materials and methods

This study was approved by the scientific and ethical committees of collaborating hospitals and the Oxford University Tropical Research Ethical Committee (NCT01421732) and The Research Ethics Review Committee of Queen Sirikit National Institute of Child Health (QSNICH) (REC.082/2562). This study followed the Transparent Reporting of a Multivariable Prediction Model for Individual Prognosis or Diagnosis (TRIPOD) guidelines for prediction model development [28].

### Data resource

We developed risk prediction models using two existing datasets from (i) a prospective cohort study of children with dengue conducted by researchers at the Oxford University Clinical Research Unit (OUCRU) in Ho Chi Minh City with seven collaborator hospitals in Southern Vietnam between 2010 and 2013 and (ii) a prospective cohort study of children with dengue conducted at the QSNICH in Bangkok, Thailand. We used dataset (i) as the training dataset and (ii) as the external validation dataset, as the Thai dataset contained too few outcome events for both endpoints to support robust model development without risking overfitting and unstable estimates. A summary of the data collection process is provided in the S1 Text pp 1–2, and full details are available in Nguyen et al. [21] and Kalayanarooj et al. [29].

### Prediction outcome

This study considered two clinical endpoints: DSS as the primary endpoint and moderate plasma leakage and/or DSS as a secondary endpoint. DSS was defined based on pulse pressure or hypotension with signs of poor perfusion, while moderate plasma leakage was determined by hematocrit changes or imaging evidence of plasma leakage (see S1 Text pp 2 for outcome definitions and measurement). The combined endpoint identifies patients with less severe disease requiring close monitoring and prompt medical intervention in hospitals, given that DSS alone was rare and represents extreme physiological derangement. In the Vietnamese dataset, all non-hospitalized patients were assumed not to have developed either endpoint.

### Candidate predictors and predictor selection

Candidate predictors were selected based on our previous systematic review and meta-analysis [22], discussion with dengue experts (SY, SK, and PS), and data availability in the cohort. The number of candidate predictors was considered according to the 10–15 events per variable rule [30,31]. Two logistic regression models were developed: the first used predefined predictors based on WHO warning signs (vomiting, mucosal bleeding, abdominal pain or tenderness, clinical fluid accumulation, lethargy or restlessness, and platelet count). In the second model, Lasso regression was used to select the

most relevant variables. We included the predictors with the highest model accuracy and stability in the final model [32]. Model accuracy was evaluated using the regularization parameter (λ) to minimize misclassification error. Stratified 10-fold cross-validation determined the minimal λ using the Glmnet package [33]. Model stability and variable selection were performed using 1,000 bootstrap samples. All candidate predictors were included in the machine learning risk prediction models. See S1 Text pp 2 for handling of outliers and missing data.

## Model development

We aimed to predict the probability of developing DSS (primary endpoint) and the combined endpoint of moderate plasma leakage and/or DSS (secondary endpoint) using six different models: a logistic regression model using the WHO warning signs parameters as predictors (the reference model), a multivariable logistic regression model included all predictors selected by Lasso regression (see Results, Selected Predictors), and four machine learning-based models (Random forest [RF], Extreme gradient boosted tree classification [XGB], Support vector machine [SVM], and Artificial neural network [ANN]) developed using all candidate predictors. See S1 Text pp 3 for further details on the machine learning models. Logistic regression and Lasso regression were applied using the Glmnet package [33]. RF, SVM, and XGB were developed using the Caret package [34] and ANN models were implemented using the Keras package [35] in R [36]. All continuous predictors, including age and laboratory variables, were standardized for the SVM and ANN algorithms. Platelet count and aspartate aminotransferase (AST) were log-transformed (with natural base) due to the right long tail distribution.

Fig 1 summarizes the conceptual framework of model development and internal validation. The dataset was split into 45 random blocks, stratified by outcome. Each model was trained and validated on 44 blocks simultaneously, leaving one block out in turn. 10-fold cross-validation was used to tune hyperparameters, and 10-fold calibration to calibrate the model. Model training and validation were repeated 45 times to obtain mean and 95% CI estimates of predictive performance using Block Jackknife estimation.

## Model validation

**Internal validation.** We assessed the predictive performance of the models with 10-fold cross-validation and an overall measurement (the Brier score), a discrimination measurement (the area under receiver operating characteristic curves, AUROC), calibration measurements (calibration plots, calibration in the large, CITL, and calibration slope), and decision curve analysis. We also assessed sensitivity, specificity, positive predictive value (PPV), and negative predictive value (NPV). We estimated the mean and CI of the predictive performance metrics using the Block Jackknife technique (Fig 1). Calibration and decision curve analyses were performed on 10-fold cross-validation.

The threshold probability for classification was defined using the minimum distance from the left-upper corner approach, to minimize the false positive and false negative misclassification. In a sensitivity analysis, we redefined the threshold probability for classification, considering the cost of a false negative and disease prevalence, reflecting that a false negative is a far worse outcome from a clinical perspective:

$$OT = \min\left((1 - sens)^2 + r\,(1 - spec)^2\right)$$

Where $r = \frac{cost \cdot prev}{1 - prev}$. We varied the cost value from 2-100. We used the same candidate predictors and performance metrics adopted in the main analysis and predicted DSS using the two best-fitting models (Lasso regression and XGB).

In a subgroup analysis, hospitalized cases with moderate plasma leakage and/or DSS were evaluated. Logistic regression with Lasso selection, RF, SVM, and XGB models were developed and validated, and their performance was assessed using the same criteria as the main analysis.

**External validation.** The models were externally validated on the Thai dataset. We quantified the degree of relatedness between the Vietnamese (training) and Thai (external validation) datasets using the AUROC of membership

and the mean and standard deviation of the linear predictors obtained from the training set and external validation set, as suggested by Debray et al. [37]. Predictive performance on the external validation was assessed using the Brier scores, AUROCs, and calibration measurements (reliability diagrams, CITL, and calibration slope).

## Supporting information

**S1 Fig. Flow chart describing patients' outcomes in the Vietnamese training (A) and Thai external validation datasets (B).**
(DOCX)

**S2 Fig. Results of difference in means with 95% CI (the y axis) and relative difference with 95% CI in standard deviation (the x axis) of predicted logarithmic odds on the training (Left) and validation sets (Right).** The vertical and horizontal dotted lines reflect no relative difference in standard deviation and no difference in means of predicted logarithmic odds between the training and validation sets. LR: models with logistic regression and lasso selection; RF: models with random forest; XGB: models with extreme gradience boosted tree; SVM: models with support vector machine; ANN: models with artificial neural networks with 2 hidden layers.
(DOCX)

**S5 Fig. Predicted probability of dengue shock syndrome (DSS) by day of illness, age, hematocrit, white blood cell count, platelet count, lymphocyte count, albumin, and AST, stratified by country (Thailand vs. Vietnam).**
(DOCX)

**S6 Fig. Predicted probability of plasma leakage by day of illness, age, hematocrit, white blood cell count, platelet count, lymphocyte count, albumin, and AST, stratified by country (Thailand vs. Vietnam).**
(DOCX)

**S1 Table. Characteristics and clinical variables of patients with laboratory confirmed dengue infection developing dengue shock syndrome (DSS) in the training (Vietnamese) and validation (Thai) datasets.**
(DOCX)

**S2 Table. Characteristics and clinical variables of patients with laboratory confirmed dengue infection developing the combined endpoint of moderate plasma leakage and/or dengue shock syndrome in the training (Vietnamese) and validation (Thai) datasets.**
(DOCX)

**S3 Table. Considered candidate predictors of Dengue Shock Syndrome (DSS) and the combined endpoint of moderate plasma leakage and/or DSS in the Vietnamese dataset.** Predictors included in each analysis are denoted by +.
(DOCX)

**S4 Table. Global model, model selected by lasso selection with the minimum lambda and bootstraps-derived quantities for assessing the uncertainty of model.**
(DOCX)

**S5 Table. Top 10 model with most selected frequencies by a 1,000-bootstrap resampling technique.**
(DOCX)

**S6 Table. Global model, model selected by lasso selection with the minimum lambda and bootstraps-derived quantities for assessing model's uncertainty for a combined endpoint of moderate plasma leakage or DSS.**
(DOCX)

**S7 Table. Top 10 model with most selected frequencies by 1,000 bootstrap resampling techniques for a combined endpoint of moderate plasma leakage or DSS.**
(DOCX)

**S8 Table. Summary of models developed using multivariable logistic regression with Lasso selection on the training set.**
(DOCX)

**S9 Table. Optimal hyperparameters selected using Bayesian Global Optimisation with Gaussian Processes of models for dengue shock syndrome.**
(DOCX)

**S10 Table. Predictive performance of the risk prediction models for the combined endpoint of moderate plasma leakage and/or DSS in a subgroup analysis trained on hospitalised patients in the Vietnamese dataset on internal validation using 10-fold cross validation.**
(DOCX)

**S11 Table. Dengue shock syndrome predictive performance of the logistic regression with lasso selection and extreme gradient boosted tree risk prediction models trained on the Vietnamese dataset, using different cost values of a false negative results.**
(DOCX)

**S12 Table. Lower and upper bounds of hyperparameters used in each machine learning algorithms.**
(DOCX)

**S13 Table. Dengue shock syndrome predictive performance of the logistic regression with lasso selection and extreme gradient boosted tree risk prediction models trained on the Vietnamese dataset, using different cost values of a false negative results. CI: Confidence interval.**
(DOCX)

**S1 Text. Supplementary Methods.**
(DOCX)

## Acknowledgments

The authors acknowledge the research teams at the Oxford University Clinical Research Unit (OUCRU), the seven participating hospitals in Vietnam, the Queen Sirikit National Institute of Child Health (QSNICH), and the Armed Forces Research Institute of Medical Sciences (AFRIMS) for conducting the prospective cohort studies in Vietnam and Thailand and for providing the data used in this study.

Material has been reviewed by the Walter Reed Army Institute of Research. There is no objection to its presentation and/or publication. The opinions or assertions contained herein are the private views of the authors and are not to be construed as official or as reflecting the views of the Department of the Army or the Department of Defense.

## Author contributions

**Conceptualization:** Sorawat Sangkaew, Damien K Ming, Bernard Hernandez, Pau Herrero, Siripen Kalayanarooj, Anon Srikiatkhachorn, Alan L. Rothman, Nguyen Lam Vuong, Bridget Wills, Cameron Simmons, Christl A. Donnelly, Sophie Yacoub, Alison Holmes, Ilaria Dorigatti.

**Data curation:** Piyarat Suntarattiwong, Siripen Kalayanarooj, Anon Srikiatkhachorn, Darunee Buddhari, Nguyen Lam Vuong, Minh Tuan Nguyen, Sophie Yacoub.

**Formal analysis:** Sorawat Sangkaew, Bethan Cracknell Daniels, Ilaria Dorigatti.

**Methodology:** Sorawat Sangkaew, Bethan Cracknell Daniels, Damien K Ming, Bernard Hernandez, Pau Herrero, Anon Srikiatkhachorn, Alan L. Rothman, Nguyen Lam Vuong, Phung Khanh Lam, Christl A. Donnelly, Sophie Yacoub, Ilaria Dorigatti.

**Resources:** Siripen Kalayanarooj, Anon Srikiatkhachorn, Sophie Yacoub.

**Supervision:** Siripen Kalayanarooj, Alan L. Rothman, Bridget Wills, Cameron Simmons, Christl A. Donnelly, Sophie Yacoub, Alison Holmes, Ilaria Dorigatti.

**Validation:** Sorawat Sangkaew, Bethan Cracknell Daniels, Damien K Ming, Ilaria Dorigatti.

**Visualization:** Ilaria Dorigatti.

**Writing – original draft:** Sorawat Sangkaew, Bethan Cracknell Daniels, Ilaria Dorigatti.

**Writing – review & editing:** Bethan Cracknell Daniels, Damien K Ming, Bernard Hernandez, Pau Herrero, Piyarat Suntarattiwong, Siripen Kalayanarooj, Anon Srikiatkhachorn, Alan L. Rothman, Darunee Buddhari, Nguyen Lam Vuong, Phung Khanh Lam, Minh Tuan Nguyen, Cameron Simmons, Christl A. Donnelly, Sophie Yacoub, Alison Holmes.

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
