## [Decision Letter · Decision Letter 0]

25 Aug 2025

Response to Reviewers
Revised Manuscript with Track Changes
Manuscript
**Journal Requirements:**
**Additional Editor Comments (if provided):**

Dear Dr Sangkaew and colleagues,

Thank you for submitting the manuscript "Early individualized risk prediction using clinical data for children during the febrile phase of dengue in outpatient settings in Vietnam and Thailand." All three reviewers agree that the study is methodologically sound and clinically relevant; they recommend publication once a short list of clarifications and expansions is addressed. I therefore invite you to submit a revised manuscript. Please provide a clean copy, a tracked-changes version, and a detailed, point-by-point response.

Key issues raised by the reviewers and the actions required

1. Dataset split and cross-country generalizability: Both Reviewer 1 and Reviewer 3 ask why the Vietnamese cohort was used for training and the Thai cohort for external validation. Either add the reciprocal analysis (train on Thailand, validate on Vietnam) or supply a clear methodological justification (e.g., events-per-variable constraints, class imbalance). If you run the reciprocal analysis, report AUROC, calibration intercept and slope, and comment on any differences in predictor selection.

2. Performance in the Thai cohort and triage implications

+ Expand the Results/Discussion to explain why most models underestimated risk in Thailand while the ANN performed acceptably.

+ Quantify, in practical terms, how many Thai children would avoid unnecessary admission if the optimal Lasso model were applied, and how many at-risk cases it would still capture.

3. Time-to-event considerations: Reviewer 1 asks whether the models can predict when DSS will occur. State explicitly that the present work predicts risk rather than timing, then briefly discuss how survival or discrete-time hazard models could be explored in future studies.

4. Influence of ‘days to hospital’ and other cohort differences: The Thai participants presented a day later on average. Report whether adding “illness day at presentation” to exploratory models altered discrimination or calibration, and explain your final decision to include or exclude this variable.

5. Additional performance metrics: Reviewers expect to see sensitivity, specificity and 95 % confidence intervals. Add a short Methods paragraph explaining why you led with Brier score, AUROC and calibration, then supply sensitivity and specificity in the main performance text or in a supplementary table.

6. Epidemiological context and prior literature

+ Reviewer 2 requests up-to-date burden estimates. Add recent incidence figures for Vietnam and Thailand to the Introduction to underscore public-health need.

+ Cite and contrast your approach with key outpatient or early-phase models (e.g., Tan et al. 2020; Phakhounthong et al. 2018; Lee et al. 2009). Highlight how your external validation and decision-curve analysis move the field forward.

7. Definition of secondary infection: Clarify, in the Methods, how primary versus secondary dengue was ascertained (e.g., IgM/IgG ratio, PRNT). This was queried by Reviewer 2.

8. Temporal validity of older data: The Thai cohort dates to 1994-2008 and the Vietnamese cohort to 2010-2013. Reviewer 3 asks whether rising patient age, obesity prevalence and liver enzyme patterns might limit transportability. Add a closing paragraph in the Discussion on secular trends and the need for prospective validation in contemporary cohorts.

9. Editorial and presentation points

+ Rewrite sentences that begin with numerals; observe journal style for abbreviations and US spelling.

+ Re-label Supplementary Table 1 so that the “Overall” column is unambiguous.

**Reviewers' Comments:**

**Comments to the Author**

1. Does this manuscript meet PLOS Digital Health’s publication criteria?

Reviewer #1: Yes

Reviewer #2: Yes

Reviewer #3: Yes

2. Has the statistical analysis been performed appropriately and rigorously?

Reviewer #1: N/A

Reviewer #2: Yes

Reviewer #3: Yes

3. Have the authors made all data underlying the findings in their manuscript fully available (please refer to the Data Availability Statement at the start of the manuscript PDF file)?

Reviewer #1: Yes

Reviewer #2: No

Reviewer #3: Yes

4. Is the manuscript presented in an intelligible fashion and written in standard English?

Reviewer #1: Yes

Reviewer #2: Yes

Reviewer #3: Yes

Reviewer #1: This study applied six models, statistical and machine learning algorithms to the dengue outpatient data in Vietnam and validated externally with data from Thailand to predict dengue shock syndrome (DSS) and moderate plasma leakage and/or DSS, which was called the combined endpoint. The performance of all models was compared with the reference model which was logistic regression with the WHO warning signs. Logistic regression with variables selected by Lasso selection outperformed the WHO algorithm. Then the authors varied cost of failing to detect severe dengue to show the trade-off between minimising false negative and false positive.

Overall, I think the study was well written and clear. The models and data were well structured and suitable. I only have some points that would like to get further clarifications if possible.

1) The models were trained and tested by the Vietnamese data, which makes sense as the data was richer compared with the Thai data set. I do wonder what the outputs might be if the models were trained and tested independently using the Thai dengue data, whether the same set of variables will be identified as the predictors of severe dengue.

2) The variables considered in the model were those that are supposed to be available as a standard measure in outpatient setting. The findings of this study could be practically useful for patient management. I wonder if it could be demonstrated that based on the optimal model achieved in this study, not all the Thai cohort needing hospitalization? Although my understanding was that only the ANN model was well performed on the external validation. All other models seemed to underestimate the risk of severe dengue among the Thai cases. Can you comment on this?

3) One thing that I would like to see in these predictive models is the sense of time to the DSS from patient arriving at hospital. I am not sure if these algorithms could say more on the likely time of plasma leakage and/or DSS e.g. within 12 hrs, 3 days or 7 days from admission etc. This could make the patient management even more efficient, I think.

4) Although both Vietnam and Thailand are low resource settings, the data (Supplementary table 1 and 2) showed a difference of a day to reach the hospital. This factor was somehow not made it in the models which I wonder if it affected the lower risk predicted in the Thai cohort. What do the authors think about this?

5) I am not sure why there was little discussion on sensitivity and specificity of these models, perhaps there is a good explanation. I think people usually expect to learn the performance of predictive models through the accuracy, sensitivity and specificity and even some discussions of false positive and false negative. I don’t see much in this study. It would be helpful to say more on other model performance indicators what they are and why they are more appropriate to look at i.e. Brier score, AUROC, Calibration intercept and slope.

Reviewer #2: This study is a valuable and potentially publishable contribution, particularly in improving early risk stratification in outpatient dengue care. The focus on outpatient populations addresses an important gap in current clinical management, and the findings could support more efficient use of hospital resources in dengue-endemic settings.

The manuscript lacks sufficient background on the prevalence and burden of dengue in Vietnam and Thailand. Including recent epidemiological data would strengthen the rationale for the study and highlight its public health significance.

Existing prediction models for dengue severity should be referenced more thoroughly. Some important models are either not cited or not adequately discussed.

The discussion section should be strengthened with a more detailed comparison between the current model and existing models in terms of design, predictor variables, performance metrics, and clinical applicability.

It is unclear how secondary dengue infections were measured. The methods should clearly describe how this variable was defined and identified.

Sentences should begin with words rather than numerals. This applies throughout the manuscript and will improve readability and professionalism.

Reviewer #3: The manuscript is very well written with careful explaining the rationales to do the study, the methods, results and discussion. The manuscript includes many tables and nice figures which are very informative for the readers to understand the data set and the methods. Rarely research centers in Southeast Asia had a collaboration to get a big data which this study can do. I appreciate the research team and thanks all authors for writing up this paper to provide a potential tool to apply in the future to triage the dengue patients with higher risk of severe disease during the febrile phase. It is very important for countries with high burden of dengue, especially during the dengue season when the healthcare systems are overwhelmed.

However, I have several questions for this paper:

1. Why did you choose Vietnamese dataset for training and Thai dataset for validation? What will happen if you do the reverse way (Thai dataset for training and Vietnamese dataset for validation)?

2. How do differences in patient’s characteristics of the two data sets affect to the models results?

3. The data set used in this study was from the past (Thai data was from 1994-2008, Vietnamese data was from 2010-2013). Clinical characteristics of severe dengue patients have changed over time (recent observational data showed that the average age increases and there are higher proportion of obese patients, higher levels of liver transaminase…). Do you think the trained models using the data set in the past would still work well for data in present? Could you discuss more about this issue in the last paragraph of the discussion?

4. Could you please review the supplementary Table 1 and reformat it, I think the second column “overall” belongs to the Vietnamese dataset.

**Do you want your identity to be public for this peer review?** For information about this choice, including consent withdrawal, please see our Privacy Policy

Reviewer #1: No

Reviewer #2: No

Reviewer #3: No

**Figure resubmission:****Reproducibility:**To enhance the reproducibility of your results, we recommend that authors of applicable studies deposit laboratory protocols in protocols.io, where a protocol can be assigned its own identifier (DOI) such that it can be cited independently in the future. Additionally, PLOS ONE offers an option to publish peer-reviewed clinical study protocols. Read more information on sharing protocols at https://plos.org/protocols?utm_medium=editorial-email&utm_source=authorletters&utm_campaign=protocols?

---

## [Decision Letter · Decision Letter 1]

12 Dec 2025

Early individualized risk prediction using clinical data for children during the febrile phase of dengue in outpatient settings in Vietnam and Thailand

PDIG-D-25-00314R1

Dear Dr Sangkaew,

We are pleased to inform you that your manuscript 'Early individualized risk prediction using clinical data for children during the febrile phase of dengue in outpatient settings in Vietnam and Thailand' has been provisionally accepted for publication in PLOS Digital Health.

Best regards,

Phat Kim Huynh, Ph.D.

Guest Editor

PLOS Digital Health

**Additional Editor Comments (if provided):**

**Reviewer Comments (if any, and for reference):**

Reviewer's Responses to Questions

**Comments to the Author**

Reviewer #1: All comments have been addressed

Reviewer #2: All comments have been addressed

Reviewer #3: All comments have been addressed

Reviewer #4: All comments have been addressed

publication criteria?

Reviewer #1: Yes

Reviewer #2: Yes

Reviewer #3: Yes

Reviewer #4: Yes

3. Has the statistical analysis been performed appropriately and rigorously?

Reviewer #1: Yes

Reviewer #2: Yes

Reviewer #3: Yes

Reviewer #4: Yes

4. Have the authors made all data underlying the findings in their manuscript fully available (please refer to the Data Availability Statement at the start of the manuscript PDF file)?

Reviewer #1: (No Response)

Reviewer #2: No

Reviewer #3: Yes

Reviewer #4: No

5. Is the manuscript presented in an intelligible fashion and written in standard English?

Reviewer #1: Yes

Reviewer #2: Yes

Reviewer #3: Yes

Reviewer #4: Yes

Reviewer #1: I am happy with all responses.

Reviewer #2: The comments I previously raised are addressed here.

Reviewer #3: All comments have been addressed with clear explanations in the response to reviewers and additional texts in the revised manuscript.

Reviewer #4: The authors provided detailed explanations and made appropriate changes in response to the comments from the editors and reviewers. They also performed further analyses as suggested which strengthen the manuscript.

**Do you want your identity to be public for this peer review?** For information about this choice, including consent withdrawal, please see our Privacy Policy

Reviewer #1: No

Reviewer #2: No

Reviewer #3: No

Reviewer #4: No
